# Achievement of the Targets of the 20-Year Infancy-Onset Dietary Intervention—Association with Metabolic Profile from Childhood to Adulthood

**DOI:** 10.3390/nu13020533

**Published:** 2021-02-06

**Authors:** Miia Lehtovirta, Laurie A. Matthews, Tomi T. Laitinen, Joel Nuotio, Harri Niinikoski, Suvi P. Rovio, Hanna Lagström, Jorma S. A. Viikari, Tapani Rönnemaa, Antti Jula, Mika Ala-Korpela, Olli T. Raitakari, Katja Pahkala

**Affiliations:** 1Research Centre of Applied and Preventive Cardiovascular Medicine, University of Turku, 20520 Turku, Finland; lauriematthews613@gmail.com (L.A.M.); tomi.laitinen@utu.fi (T.T.L.); joel.nuotio@utu.fi (J.N.); suvrov@utu.fi (S.P.R.); olli.raitakari@utu.fi (O.T.R.); katpah@utu.fi (K.P.); 2Centre for Population Health Research, Turku University Hospital, University of Turku, 20520 Turku, Finland; hanna.lagstrom@utu.fi; 3Paavo Nurmi Centre, Sports & Exercise Medicine Unit, Department of Physical Activity and Health, University of Turku, 20520 Turku, Finland; 4Heart Center, Turku University Hospital, University of Turku, 20520 Turku, Finland; 5Department of Pediatrics and Adolescent Medicine, Turku University Hospital, University of Turku, 20520 Turku, Finland; hjniin@utu.fi; 6Department of Public Health, Turku University Hospital, University of Turku, 20520 Turku, Finland; 7Division of Medicine, Department of Medicine, Turku University Hospital, University of Turku, 20520 Turku, Finland; jorvii@utu.fi (J.S.A.V.); erkron@utu.fi (T.R.); 8Department of Chronic Disease Prevention, Institute for Health and Welfare, 20750 Turku, Finland; Antti.Jula@thl.fi; 9Computational Medicine, Faculty of Medicine, University of Oulu & Biocenter Oulu, 90014 Oulu, Finland; mika.ala-korpela@oulu.fi; 10NMR Metabolomics Laboratory, School of Pharmacy, University of Eastern Finland, 70210 Kuopio, Finland; 11Department of Clinical Physiology and Nuclear Medicine, Turku University Hospital, University of Turku, 20520 Turku, Finland

**Keywords:** diet, quality of diet, metabolomics, metabolic profiling, primordial prevention

## Abstract

The Special Turku Coronary Risk Factor Intervention Project (STRIP) is a prospective infancy-onset randomized dietary intervention trial targeting dietary fat quality and cholesterol intake, and favoring consumption of vegetables, fruit, and whole-grains. Diet (food records) and circulating metabolites were studied at six time points between the ages of 9–19 years (*n* = 549–338). Dietary targets for this study were defined as (1) the ratio of saturated fat (SAFA) to monounsaturated and polyunsaturated fatty acids (MUFA + PUFA) < 1:2, (2) intake of SAFA < 10% of total energy intake, (3) fiber intake ≥ 80th age-specific percentile, and (4) sucrose intake ≤ 20th age-specific percentile. Metabolic biomarkers were quantified by high-throughput nuclear magnetic resonance metabolomics. Better adherence to the dietary targets, regardless of study group allocation, was assoiated with higher serum proportion of PUFAs, lower serum proportion of SAFAs, and a higher degree of unsaturation of fatty acids. Achieving ≥ 1 dietary target resulted in higher low-density lipoprotein (LDL) particle size, lower circulating LDL subclass lipid concentrations, and lower circulating lipid concentrations in medium and small high-density lipoprotein subclasses compared to meeting 0 targets. Attaining more dietary targets (≥2) was associated with a tendency to lower lipid concentrations of intermediate-density lipoprotein and very low-density lipoprotein subclasses. Thus, adherence to dietary targets is favorably associated with multiple circulating fatty acids and lipoprotein subclass lipid concentrations, indicative of better cardio-metabolic health.

## 1. Introduction

Early exposure to biological, behavioral, and environmental risk factors can increase the possibility of cardio-metabolic disease in adulthood [1,2,3]. Thus, the primordial prevention of cardio-metabolic diseases beginning in childhood is warranted [4]. Diet is an important target, since it influences several known cardio-metabolic risk factors, such as adiposity, glucose regulation, and serum lipids [5].

Metabolomics is an important component in the study of disease pathophysiology and identification of biomarkers of disease risk [6,7,8,9]. High-throughput serum nuclear magnetic resonance (NMR) metabolomics allows the quantification of e.g., lipoprotein subclasses, amino acids, and metabolites of glycolysis [6]. There is a paucity of data on the effects of dietary modification on the metabolic profile of children throughout their development.

The Special Turku Coronary Risk Factor Intervention Project (STRIP) was established in 1989 to investigate the effect of family-based dietary intervention sustained from infancy to early adulthood on cardio-metabolic risk factors [10]. The dietary counselling intervention reflected Nordic Nutrition Recommendations [11], and specifically targeted the quality of dietary fat and cholesterol intake, with complementary counselling on the consumption of fruit, vegetables, and whole grains. Previous reports have shown that the STRIP intervention is effective in modifying children’s diet [12,13,14], and is associated with numerous beneficial cardio-metabolic outcomes [12,15,16,17,18,19].

A prior study about the STRIP intervention effect on a comprehensive metabolic profile showed that the intervention led to favorable changes in circulating fatty acids and lipoprotein subclass lipids [20]. We also recently examined the effects of achieving the STRIP intervention dietary targets, regardless of study group allocation, on insulin sensitivity and serum lipids [21]. In the study, having a diet with fat quality at the target level and favorable fiber and sucrose intake was associated with lower concentrations of serum glucose, insulin, and conventional lipid measures. Furthermore, the study revealed that there were participants in both the intervention and control group that had diets that met the targets for dietary fat quality, fiber, and sucrose intake. Thus, achieving these dietary targets applies to both subgroups in the study and the effect of achieving the dietary targets on the comprehensive metabolic profile, regardless of study group, remains unknown. Building upon these findings, this study provides novel data on how attaining dietary targets in keeping with dietary recommendations is reflected in a detailed metabolic profile measured repeatedly between ages 9 and 19 years.

## 2. Materials and Methods

### 2.1. Study Design and Participants

The STRIP study is a prospective, randomized intervention trial targeting atherosclerosis risk factors beginning in infancy (age seven months) and ending in early adulthood (age 20 years). The study was conducted between 1990 and 2011; recruitment protocol, dietary intervention, and data collection methods have been elaborated previously [10]. Briefly, families with healthy infants living in Turku, Finland were recruited by nurses at well-baby clinics during routine five-month visits between February 1990 and June 1992 (*n* = 1880). Of those eligible, 56.7% (*n* = 1062; Caucasian) participated and were randomized either to the intervention (*n* = 540; 256 girls) or control (*n* = 522; 256 girls) group at age seven months. Both groups attended study visits led by a nutritionist and a pediatrician or nurse.

The STRIP intervention group visited the study center for individualized dietary counseling and nutrition education sessions at one- to three-month intervals until the child was two years old, and biannually thereafter until 20 years of age [10,22]. The main dietary goal was to reduce the intake of SAFAs and increase the unsaturated to saturated fat ratio. Additionally, low intake of dietary cholesterol, sodium, and sucrose were targeted, coupled with enhancing the consumption of whole-grain products, fruit, and vegetables. Recommendations were based on the latest available Nordic Nutrition Recommendations [11]. The optimal diet was comprised of energy without restrictions, with 10–15% of all energy (E%) from protein, 50–60% from carbohydrates, and up to 30% from fat, except between ages one and two years, where 30–35% energy from fat was recommended. A fixed diet was never demanded, rather, dietary changes were suggested based on the child’s food records.

The control group had biannual visits until the child was seven years of age, after which families were seen annually until the child was 20 years old. Similar measurements were performed for both study groups, and they met the same study personnel. Control families did not receive any dietary intervention and were given basic health education routinely given at Finnish well-baby clinics and by school healthcare.

For the present analyses, data were obtained from children reporting dietary data and having metabolic biomarkers quantified by high-throughput NMR metabolomics, at the same time point and in at least one of the six time points at age nine (*n* = 549, 47% intervention), 11 (*n* = 518, 46%), 13 (*n* = 485, 45%), 15 (*n* = 472, 44%), 17 (*n* = 410, 42%), or 19 (*n* = 338, 41%) years, representing 71–85% of the total study participants.

The STRIP study has been approved by the Ethics Committee of Turku University and Turku University Central Hospital. Written informed consent was obtained from all families at the beginning of the study and from adolescents at 15 and 18 years of age. The project identification code/Clinical Trial Registration for the study is: ClinicalTrials.gov (accessed on 6 February 2021), STRIP19902010, unique identifier: NCT00223600, http://www.clinicaltrials.gov (accessed on 6 February 2021).

### 2.2. Dietary Data and Dietary Target Score

Food consumption was recorded using a three-day food record at ages eight, 13, and 18 months, while a four-day food record was applied from ages two to 20 years (consecutive days; at least one weekend day included) [10,22]. Parents or other caregivers (i.e., nanny, etc.) were responsible for filling out the food record during early childhood. When the child began attending daycare, and subsequently school, the establishment’s personnel were also asked to assist the child in completing the food records. As the children aged, they were given more responsibility in completing the food records, however, parents were still advised to check the records and assist the child. Throughout the study, the child and the parents/caregivers were given written and verbal instructions on how to fill out the food record. A detailed food picture booklet was used to assist in the estimation of the amounts of food/drink beginning from the age of 13 years. A dietitian checked the food records for completeness and accuracy during the study visit and, if necessary, added missing details after discussion with the child/parents. The food and nutrient intakes were analyzed with a Micro-Nutrica^®^ program (Research Center of the Social Insurance Institution, Turku, Finland) [23]. The food and nutrient database of the program was continuously updated based on the foods/drinks reported by the participants. This allowed the production of detailed dietary data, simultaneous updates related to changes in available foods over the years, and an accurate reflection of the foods/drinks consumed by the age group.

The dietary target score was created to reflect the achievement of key STRIP dietary intervention goals [21]. The quality of dietary fat was assessed using two separate targets: the ratio of SAFAs to monounsaturated and polyunsaturated fatty acids (MUFA + PUFA) < 1:2, and as the intake of SAFA < 10 percentage of energy intake (E%). Fiber intake reflected whole-grain product, fruit, and vegetable consumption, which were secondary goals of the intervention. The dietary fiber target was defined as being at the top age-specific quintile (≥80th age-specific percentile). This criterion was chosen because very few participants met the goal set by Nordic Nutrition Recommendations (≥3 g/MJ) [21,24]. As a further reflection of the dietary carbohydrate quality, the desired sucrose intake (E%) was defined as being in the lowest age-specific quintile (≤20th age-specific percentile). This criterion was used in the absence of a consensus recommendation on sucrose intake.

To form the dietary target score, in every time point, participants were given one point for meeting each of the four targets: (1) SAFA/(MUFA + PUFA) < 1:2, (2) SAFA < 10% of energy, (3) dietary fiber ≥ 80th age-specific percentile, and (4) sucrose ≤ 20th age-specific percentile. The score range was 0–4 points. For the present analyses, two binary variables indicating participants meeting 1–4 targets and participants meeting 2–4 targets were created. This approach was chosen because of the low prevalence of participants achieving three or four targets, and to study whether achieving even one of the targets was beneficial.

Furthermore, low intake of dietary cholesterol was targeted in the intervention [25]. We added an additional target for dietary cholesterol, and formed a secondary dietary target score with range of 0–5 points. The percentage of the STRIP study participants having dietary cholesterol <300 mg/d, as recommended e.g., in the Dietary Guidelines for Americans [26], ranged between 65–91% at different time points. Since the majority of study participants reached this goal, we defined being in the lowest age-specific quintile as the desired cholesterol intake.

### 2.3. Lipid and Metabolite Quantification

A high-throughput NMR metabolomics platform was used for the quantification of 60 serum lipid and metabolite measures. This metabolomics platform provides the simultaneous measurement of routine lipids and lipid concentrations of 14 lipoprotein subclasses, and further quantifies fatty acids, amino acids, ketone bodies, and gluconeogenesis related metabolites in absolute concentration units [6]. Metabolic profiling was measured from fasting serum samples at six time points collected when the study participants were aged 9 to 19 years. The NMR metabolomics platform has been used in a wide range of studies [6,8].

### 2.4. Statistical Analyses

Metabolic measures with skewed distributions were log(x + 1)-transformed prior to the analyses. The abovementioned binary target variables indicating participants meeting 1–4 targets and participants meeting 2–4 targets were used for the analyses. For the secondary dietary target score, including targeted cholesterol intake, the same criteria were applied.

Previously, the STRIP study has reported significant sex interactions in lipid-lowering effects of the intervention [12]. To study possible sex interaction with the dietary target score, a linear mixed model for each measure of metabolite concentration and dietary target score was formed, with dietary target score, age, sex, and sex*dietary target score as the fixed effects, and the subject as a random effect. Two metabolite measures showed significant interaction between sex and dietary target score. Thus, analyses with binary dietary target variables were conducted sex-combined by fitting a linear mixed-effects model for repeated measures, with dietary target variable, sex, and age as fixed effects and the subject as a random effect. To facilitate the comparison of effect sizes across metabolites, all metabolic measures were scaled to standard deviation (SD) units. Effect sizes reported hereby correspond to the average difference in SD-scaled metabolite concentration due to achieving 1–4 or 2–4 dietary targets, compared to achieving none of the dietary targets, and are from pooled analyses across the six time points. To account for 60 independent tests, *p*-values < 0.001 were denoted as significant. Statistical analyses and graphics were done using R 3.5.1 software [27], the nlme package [28] was used to perform the linear mixed effect model analyses, and figures were produced using the package ggplot2 [29].

## 3. Results

This study investigated the association of achieving ≥1 or ≥2 dietary targets compared with achieving none of the targets in the circulating metabolic profile of 338–549 participants assessed at six time points between 9 and 19 years of age. The percentage of participants achieving specific dietary targets and the percentage of them achieving 0, ≥1, or ≥2 targets at ages 9 to 19 are shown in Appendix A. The percentage of intervention and control children achieving these targets are shown in Appendix A. Appendix A includes corresponding data when cholesterol intake was also included in the dietary target score. Achieving some dietary targets generally resulted in better performance in other dietary variables, as shown in Appendix A.

### 3.1. Association of Achieving Dietary Targets with Serum Metabolic Profile

The results of achieving ≥1 or ≥2 dietary targets compared to achieving 0 targets in detailed circulating serum metabolites are shown in Figure 1, Figure 2 and Figure 3. Most associations between dietary targets met and metabolites were found in circulating fatty acids and lipoprotein lipids; generally, the more dietary targets achieved, the stronger the association with the metabolite measure. All metabolic differences between dietary target score ranges are shown in SD-scaled concentration units to enable comparison across metabolic measures. Effect estimates for the metabolic measures in absolute concentration units are listed in Appendix A.

### 3.2. Circulating Serum Fatty Acids

Figure 1 illustrates the associations between the achieved dietary targets and circulating serum fatty acid concentrations when ≥1 or ≥2 of the targets were met, compared to achieving none of the targets. Where the most prominent associations were observed, a higher number of targets met was generally associated with greater changes in serum fatty acid concentrations. Achieving more of the dietary targets was associated with an evident increase in circulating omega-3 fatty acid concentrations and total PUFAs in proportion to total fatty acids, whereas a slightly lesser increase in omega-6 fatty acids was observed. These associations were consistent with those found in circulating docosahexaenoic acid (DHA; an omega-3 fatty acid), where serum DHA was increased in a stepwise fashion with ≥1 and ≥2 dietary targets met, and the association between dietary targets and serum linoleic acid (LA; an omega-6 fatty acid) showed an increased tendency. The serum proportion of MUFAs showed a tendency towards reduced concentrations when more dietary targets were achieved. A greater number of dietary targets achieved was associated with greater reductions in the proportion of SAFAs. More dietary targets achieved was also associated with a greater degree of fatty acid unsaturation, but no association between fatty acid chain length and the achieved dietary target was found.

### 3.3. Lipids and Lipoproteins

The associations between the dietary targets met and a broad panel of lipoprotein lipid measurements are depicted in Figure 2. In most routine lipid measures, achieving more of the targets was associated with lower concentrations of total cholesterol, LDL cholesterol, and apolipoprotein B. No associations were observed between the dietary target score and HDL cholesterol. Overall, the lipoprotein subclass profiling showed a tendency towards lower lipid concentrations in IDL and VLDL lipoprotein subclasses the more dietary targets were attained. This was reflected in the reduced levels of remnant-cholesterol (=VLDL cholesterol + IDL cholesterol). There was a significant sex interaction (*p* = 0.01) observed for small VLDL subclass lipid concentration, with boys showing a tendency toward lower lipid concentration with more dietary targets achieved, whereas no association was found for girls. For LDL subclasses, achieving more dietary targets was associated with reduced lipoprotein lipid concentrations. For medium and small HDL subclasses, a reduction in lipid concentrations was also observed with more dietary targets achieved, whereas no association was found with large and extra-large HDL subclass lipid concentrations. When ≥1 or ≥2 dietary targets were achieved, an equal increase was found in the LDL particle size compared to the LDL particle size when zero dietary targets were achieved. No association was observed between the dietary targets met and VLDL or HDL particle size. Attaining more dietary targets was associated with a tendency toward lower concentrations in circulating IDL cholesterol and VLDL cholesterol.

### 3.4. Non-Lipid Metabolites

The association between achieved dietary targets and serum glycolysis related metabolites, amino acids, ketone bodies, metabolic waste products, and other non-lipid circulating metabolites are shown in Figure 3. Glucose, lactate, and pyruvate concentrations showed a tendency toward lower concentrations with an increasing number of achieved dietary targets. No association between the targets met and circulating citrate and glycerol concentrations was observed. A tendency towards lower serum concentration was observed for tyrosine as more targets were achieved. There were no associations found in other amino acids, ketone bodies, or metabolic waste products with the achieved dietary targets. Achieving the dietary targets was not associated with albumin or glycoprotein levels either, but there was a tendency to lower concentrations of phosphoglycerides, phosphatidylcholines, and sphingomyelin with more dietary targets attained. A sex interaction (*p* = 0.02) for glycoprotein was observed with girls, showing a tendency for lower concentration with more targets achieved, whereas no associations were found in boys.

### 3.5. Dietary Target Score Including Cholesterol

We also applied a secondary dietary target score, where low intake of cholesterol (≤20th age-specific percentile) was added to the score. When similar analyses to those reported in Figure 1, Figure 2 and Figure 3 were conducted, the associations observed between meeting ≥1 or ≥2 of the targets and the metabolic profile remained essentially unchanged (Appendix A). For instance, the strong associations of a reduced proportion of serum SAFAs and an increased number of double bonds per fatty acid with more dietary targets attained remained after including the cholesterol target. The trends found in the metabolic profiles of lipoprotein subclasses as more of the dietary targets were attained remained similar. In addition, Appendix A describes the associations of the dietary target score when 0, 1, 2, or ≥3 targets were achieved.

## 4. Discussion

This study demonstrates that, from childhood to early adulthood, a diet concordant to the targets of the STRIP intervention and dietary recommendations regarding fat and carbohydrate quality is associated with beneficial changes in serum metabolites, especially those relating to circulating fatty acids, which are predictive of cardio-metabolic event risk and type 2 diabetes. Furthermore, achieving a higher number of dietary fat, carbohydrate quality, and cholesterol targets is associated with more pronounced changes in circulating metabolites, including some non-lipid metabolites. Better adherence to dietary targets resulted in a higher serum proportion of PUFAs, particularly serum omega-3 fatty acid concentrations, a lower proportion of serum SAFA, reduced lipid concentrations in LDL subclasses, as well as in medium and small HDL subclasses, and increased LDL particle size. The favorable metabolic associations were, however, often evident even when just one dietary target was attained. Furthermore, when compared to the STRIP intervention effect [20], having a diet concordant to the intervention targets was associated with a more beneficial cardio-metabolic health profile.

Overall, the observed associations were towards a less risk-prone metabolic profile for cardio-metabolic disease and type 2 diabetes [30,31]. A recent study conducted on two adult population cohort studies examined metabolite profiles associated with adherence to the Alternative Healthy Eating Index [32]. It found, in concordance with our study, that a healthier diet was associated with a higher degree of unsaturation of fatty acids, and higher ratios of polyunsaturated fatty acids, omega-3, and DHA relative to total fatty acids. Another randomized trial studying the effects of wholegrain, fish, and bilberries on the metabolic profile of an adult population with metabolic syndrome for a 12-week period found that adherence to a healthier diet resulted in increased poly-unsaturation of fatty acids, and shifted HDL particle subclass distribution toward larger particles [33]. Our study shows similar results in healthy children followed from infancy to early adulthood. The unique study design of the STRIP study limits possibilities for comparison, since similar study designs are scarce. A study on young adults from the Finnish Twin Cohort Study examined the association between habitual diet and lipoprotein subclass profile [34]. It identified a “junk food” pattern, correlating positively with the energy from fat and sucrose and negatively with total carbohydrates and dietary fiber. This pattern showed associations with increased concentrations of triglycerides and smaller sized HDL particles, reduced LDL particle size, and increased VLDL particle size. In our study, the dietary targets of better fat quality and lower sucrose and higher fiber intake were in the opposite direction of the “junk food” pattern. This is reflected in the lipid and lipoprotein profile, as a reduced trend was seen in the concentration of serum triglycerides and VLDL particle size and as an increase in LDL particle size.

We have previously shown that STRIP study participants who achieved more dietary targets had lower concentrations of fasting serum glucose, insulin, HOMA-IR, LDL cholesterol, and non-HDL-cholesterol, and a lower ratio of apolipoprotein B/apolipoprotein A1 throughout the 20 years of the study [21]. These current results extend the previous findings by showing the associations of achieving more of the dietary targets on the detailed lipoprotein subclass patterns, as well as on circulating fatty acids and non-lipid metabolites. The observed tendency in VLDL subclasses and IDL lipid concentrations and significantly lower lipid concentrations in LDL subclasses is in line with a reduced risk for cardio-metabolic disease [30]. Lower lipid concentrations in the medium and small HDL subclasses were also observed. Currently, there is no consensus on the role of HDL subclasses on cardiovascular disease risk [35,36,37]. Our study also shows a tendency toward lower concentrations of glycolysis- and gluconeogenesis-related metabolites as more dietary targets are achieved. A similar tendency toward lower concentrations with more dietary targets attained is observed in branch-chain amino acids and aromatic amino acids, especially tyrosine, all of which have been linked with insulin sensitivity [31,38]. These findings on non-lipid metabolites are towards a lesser future risk of type 2 diabetes and cardio-metabolic disease [39,40].

We recently showed that the dietary intervention given in STRIP from infancy to early adulthood resulted in significant differences in the fatty acid intakes between the intervention group and the control group [20]. These differences were favorably reflected in the metabolic profiles of the intervention children. The most prominent metabolic effects were seen in the fatty acid balance, with intervention boys also displaying favorable lowering of triglyceride-rich lipoproteins. For non-lipid metabolites, the intervention effect was limited, with increased levels of glutamine as the only exception. Overall, the results were in line with lower cardio-metabolic risk. In this study, we examined the metabolic profiles of those who achieved the dietary targets of the intervention, regardless of the study group. Generally, achieving the dietary targets yielded a similar pattern of metabolic profile when compared with sex-combined results of the dietary intervention. Meeting the dietary targets was reflected especially as an increased proportion of circulating omega-3 fatty acids, number of double bonds per fatty acid, and LDL particle size when compared with receiving the dietary intervention. Those who achieved dietary targets also showed stronger reductions in total cholesterol, LDL subclass lipid concentration, and medium and small sized HDL subclass lipid concentrations. Unlike the intervention study, attaining dietary targets showed a tendency toward lower concentrations of glucose, lactate, pyruvate, and tyrosine, whereas no changes were found in the concentrations of glutamine. Thus, fulfilling the dietary targets was associated with an even more favorable metabolic profile with regards to cardio-metabolic risk than receiving the dietary intervention alone.

The strengths of our study include the uniqueness of the STRIP trial, with an aim to affect the dietary choices of the study participants for 20 years beginning in infancy, and the large number of participants with a detailed metabolic profile measured at multiple time points with reliable methods. Since this study is the first to report the association of better dietary fat quality, higher intake of fiber, and lower intake of sucrose and cholesterol with several serum metabolites using high throughput NMR in healthy children at six time points between the ages of 9 and 19 years of age, similar study designs are non-existent, and this limits possibilities for comparison. In a such a long study design, a sizable loss at follow-up is to be expected. The characteristics of those participating in the study and those lost at follow-up have been compared repeatedly, and no differences have been found regarding body weight, BMI, serum total cholesterol, or saturated fat intake [10,19,41]. Recently, we reported detailed loss-at-follow-up analyses regarding components of the metabolic syndrome and STRIP study group, and found that discontinuance in the study was not affected by these characteristics [15]. Other limitations include a potential selection bias during the recruitment phase, as more health-oriented families might have taken part in the study. We also acknowledge that the use of sample-specific cut-points for fiber, sucrose, and cholesterol intake limits the ability of an individual to achieve a high score. Furthermore, the data collection period of the study was completed in 2011, thus, the dietary data in this study may not fully reflect the diet of contemporary Finnish children and adolescents.

## 5. Conclusions

In conclusion, achieving the dietary targets was favorably reflected in the metabolic profile from childhood to early adulthood. Achieving more dietary targets enhanced the metabolic profile further, but, importantly, our findings support the notion that fulfilling even one of these dietary targets is reflected in a more favorable metabolic profile with regards to cardio-metabolic disease risk. These results thus lend evidence to the recommendations of favoring unsaturated fat e.g., in terms of using oils, consumption of fiber-rich foods, including whole-grains and vegetables, and low intake of sucrose by e.g., limiting the consumption soft drinks and candy.

## Figures and Tables

**Figure 1 nutrients-13-00533-f001:**
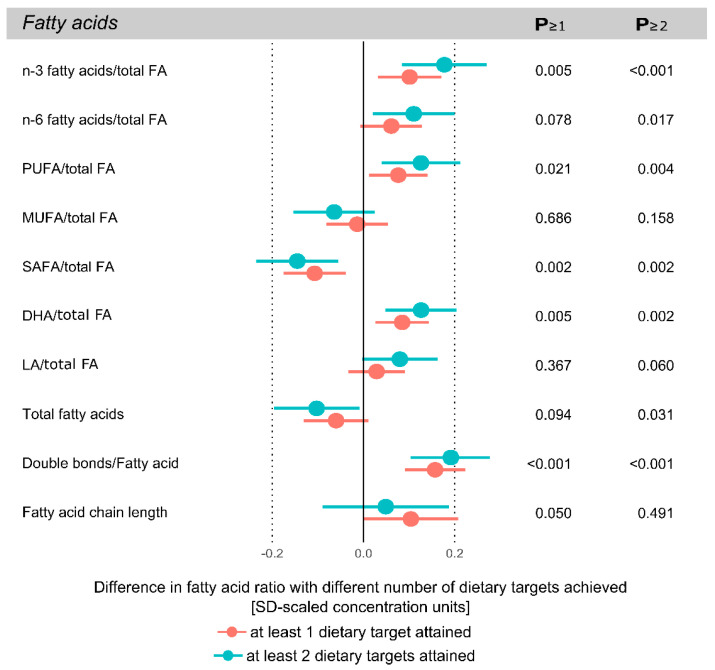
Differences in circulating serum fatty acids between achieving the dietary targets compared to achieving none of the dietary targets. Effect estimates are standard deviation scaled differences between achieving zero dietary targets with respect to achieving ≥1 (red) or ≥2 (blue) of the targets. Error bars indicate 95% confidence intervals. Metabolic measures are from pooled analyses across the six time points; n-3 fatty acids/total FA denotes the ratio of omega-3 fatty acids to total fatty acids; PUFA, polyunsaturated fatty acids; MUFA, monounsaturated fatty acids; SAFA, saturated fatty acids; DHA, docosahexaenoic acid; LA, linoleic acid.

**Figure 2 nutrients-13-00533-f002:**
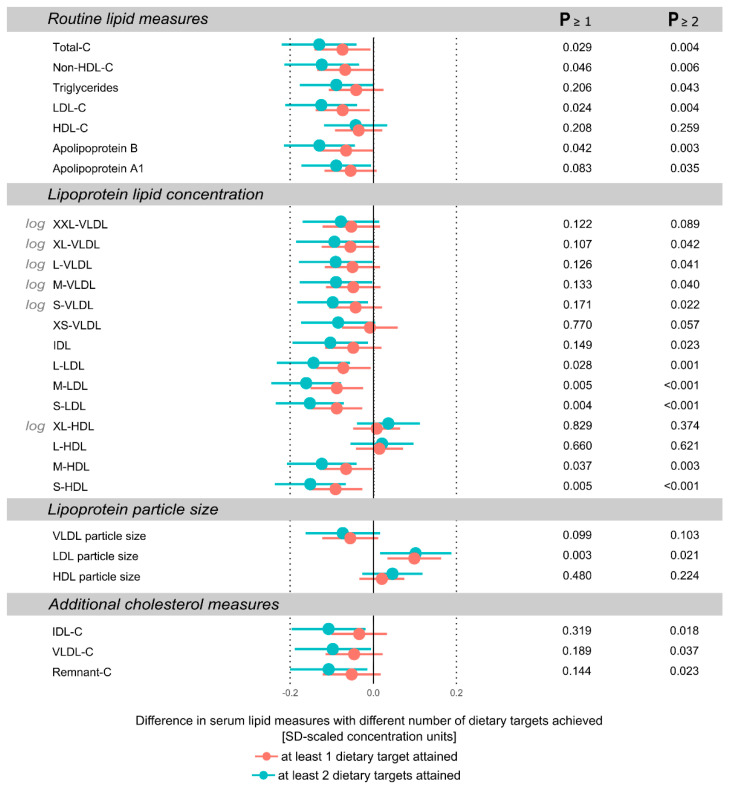
Differences in serum lipid measures between achieving the dietary targets compared to achieving none of the dietary targets. Effect estimates are standard deviation scaled differences between achieving zero dietary targets with respect to achieving ≥1 (red) or ≥2 (blue) of the targets. Error bars indicate 95% confidence intervals. Lipid measures are from pooled analyses across the six time points, and those with skewed distributions were log(x + 1)-transformed prior to analyses. C, cholesterol; HDL, high-density lipoprotein; LDL, low-density lipoprotein; VLDL, very low-density lipoprotein; IDL, intermediate-density lipoprotein.

**Figure 3 nutrients-13-00533-f003:**
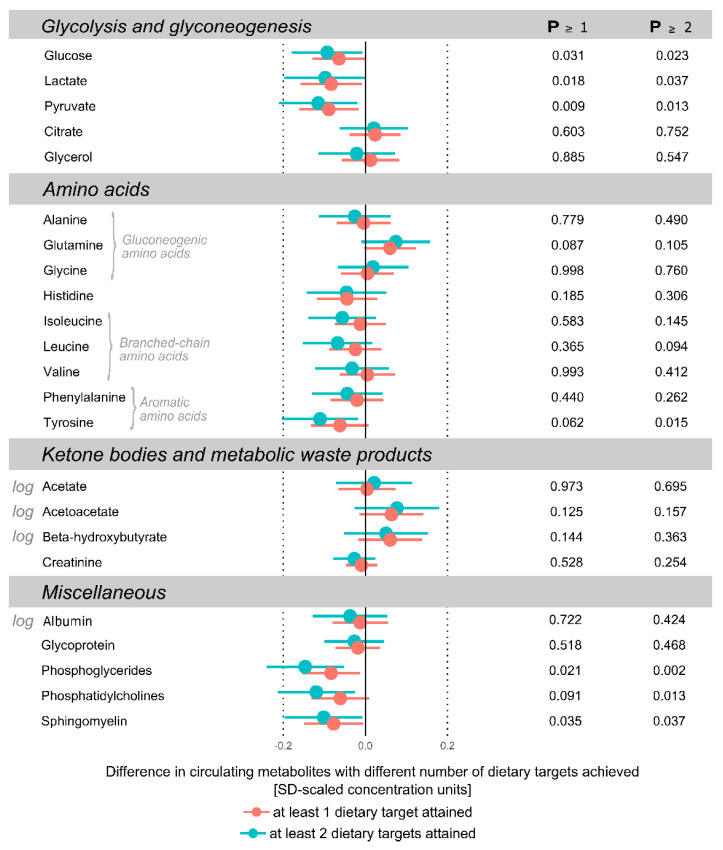
Differences in circulating metabolites between achieving the dietary targets compared to achieving none of the dietary targets. Effect estimates are standard deviation scaled differences between achieving zero dietary targets with respect to achieving ≥1 (red) or ≥2 (blue) of the targets. Error bars indicate 95% confidence intervals. Metabolic measures are from pooled analyses across the six time points, and those with skewed distributions were log(x + 1)-transformed prior to the analyses.

## Data Availability

The rights to the data belong to the STRIP research group. Selected variables and their descriptions without personal identification codes are distributed to investigators and collaborators working on specific projects. Data sharing outside the STRIP group requires a data-sharing agreement. Investigators can submit an expression of interest to the STRIP Steering Committee.

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
