# Peer review of "Achievement of the Targets of the 20-Year Infancy-Onset Dietary Intervention—Association with Metabolic Profile from Childhood to Adulthood"

_nutrients, 2021, doi:10.3390/nu13020533_

Round 1

Reviewer 1 Report

This is a well written, interesting manuscript, employing appropriate methodology that will contribute to the literature.

Reviewer 2 Report

Overall the topic is quite interesting with a good number of participants in such a long follow-up study.  Well done!

Just a few questions for the authors to clarify:

1. What's the rationale for choosing the sucrose intake

≤ 20th age-specific percentile? 

2. In Line 118, why do the author choose age 9 (not at other age) to start collecting the metabolic biomarkers? 

3. Why only looking at the associations between achieved dietary targets and circulating serum fatty acid concentrations when ≥1 or ≥2 of the targets were met compared to achieving none of the target?Why not also look at the association when ≥ 4 of the targets were met?

4, For Line 125-130, were the child, parents or caregivers given some education given by a registered dietitian regarding how to fill out the food record appropriately? How do the authors deal with self-reporting accuracy issue in this study?

5. In the Abstract section, the authors made a mistake on this sentence: "Diet (food records) and circulating metabolites were studied at six time points between ages 9-19 years (n=338-549)."  The number inside the bracket should be (n=549-338).

Reviewer 3 Report

The paper deals with a study based on a very large project focused on the prevention of coronary risk factor on young individuals from infancy to adulthood; the topic of the paper appears of great interest, and with a high degree of novelty in its field.

The data obtained from the project were analysed by the well-established metabolomic NMR technique. However, methodology analysis would have been more accurate if it had been possible to access the supplementary material referred by the authors, but not available on the website. Thus, reviewing will take into account only the results expressed in table form in the main paper.

Materials and methods

The methodology appears consistent with the aim and fits within the scope of the paper. Hypotheses formulated appear consistent, despite the dataset belongs to a study completed in 2011 and therefore probably somehow outdated; this aspect should be clarified in the text. It should be also noted that statistical analysis software is merely cited, so the section should be clarified by reporting R software routines used.

Results

The authors discuss in this section the effects of the association of dietary targets achievement in comparison with the serum metabolic profile of individuals included in the study. The assumptions are clearly explained, and tables reflect the outcome of data analysis. However, lack of access to supplementary material prevents from giving a definitive judgment on results.

Overall conclusion

The paper fits within the scope of the journal and is suitable for publication after minor changes. Paper readability could also benefit from a critical revision by a native English speaker.

Reviewer 4 Report

Dear authors,

I would like to congratulate you with this manuscript. I believe it is a high quality paper. The question that arose while reading (e.g. loss to follow up) were discussed in the paper. I have no additional comments to make.
